# T2DB: A Web Database for Long Non-Coding RNA Genes in Type II Diabetes

**DOI:** 10.3390/ncrna9030030

**Published:** 2023-05-08

**Authors:** Rebecca Distefano, Mirolyuba Ilieva, Jens Hedelund Madsen, Hideshi Ishii, Masanori Aikawa, Sarah Rennie, Shizuka Uchida

**Affiliations:** 1Section for Computational and RNA Biology, Department of Biology, University of Copenhagen, DK-2200 Copenhagen, Denmark; fwh492@alumni.ku.dk; 2Center for RNA Medicine, Department of Clinical Medicine, Aalborg University, DK-2450 Copenhagen, Denmark; mirolyubasi@dcm.aau.dk (M.I.); jenshm@dcm.aau.dk (J.H.M.); 3Center of Medical Innovation and Translational Research, Department of Medical Data Science, Graduate School of Medicine, Osaka University, Suita 565-0871, Japan; hishii@gesurg.med.osaka-u.ac.jp; 4Center for Interdisciplinary Cardiovascular Sciences, Cardiovascular Division, Brigham and Women’s Hospital, Harvard Medical School, Boston, MA 02115, USA; maikawa@rics.bwh.harvard.edu; 5Channing Division of Network Medicine, Brigham and Women’s Hospital, Harvard Medical School, Boston, MA 02115, USA; 6Center for Excellence in Vascular Biology, Brigham and Women’s Hospital, Harvard Medical School, Boston, MA 02115, USA

**Keywords:** diabetes, database, lncRNA, RNA-seq

## Abstract

Type II diabetes (T2D) is a growing health problem worldwide due to increased levels of obesity and can lead to other life-threatening diseases, such as cardiovascular and kidney diseases. As the number of individuals diagnosed with T2D rises, there is an urgent need to understand the pathogenesis of the disease in order to prevent further harm to the body caused by elevated blood glucose levels. Recent advances in long non-coding RNA (lncRNA) research may provide insights into the pathogenesis of T2D. Although lncRNAs can be readily detected in RNA sequencing (RNA-seq) data, most published datasets of T2D patients compared to healthy donors focus only on protein-coding genes, leaving lncRNAs to be undiscovered and understudied. To address this knowledge gap, we performed a secondary analysis of published RNA-seq data of T2D patients and of patients with related health complications to systematically analyze the expression changes of lncRNA genes in relation to the protein-coding genes. Since immune cells play important roles in T2D, we conducted loss-of-function experiments to provide functional data on the T2D-related lncRNA *USP30-AS1*, using an in vitro model of pro-inflammatory macrophage activation. To facilitate lncRNA research in T2D, we developed a web application, T2DB, to provide a one-stop-shop for expression profiling of protein-coding and lncRNA genes in T2D patients compared to healthy donors or subjects without T2D.

## 1. Introduction

RNA sequencing has revealed that most of the mammalian genome is transcribed as RNA [1,2], although only a few percent of annotated transcribed regions correspond to the exons of protein-coding genes, with the rest being collectively termed non-coding RNAs (ncRNAs). These ncRNAs include ribosomal RNAs (rRNAs), transfer RNAs (tRNAs), small RNAs (e.g., microRNAs (microRNAs), small nuclear RNAs (snRNAs), and small nucleolar RNAs (snoRNAs)), and long ncRNAs (lncRNAs) [3,4]. LncRNAs have widely attracted the attention of researchers during the last two decades. They loosely refer to any ncRNA exceeding 200 nucleotides (nt) in length [5,6] and are typically named and characterized based on their nearest protein-coding gene [7,8]. Due to their broad definition and cell-type-specific expression patterns, many thousands of lncRNAs have been discovered in recent years [9,10], which have been implicated in a diverse range of functions including imprinting, epigenetic, transcriptional, post-transcriptional and translation regulation, via binding to other macromolecules (i.e., DNA, RNA, and proteins) [11,12,13,14,15]. As many lncRNAs are expressed in disease, it has been speculated that lncRNAs have functional roles in the pathogenesis of various diseases, with some of these roles being validated experimentally [16,17,18,19].

Type II diabetes (T2D) affects approximately 462 million individuals worldwide, corresponding to 6.28% of the world’s population [20]. The causes of T2D include obesity, physical inactivity and genetic factors [21,22,23], and if left untreated, T2D can result in further health complications, including cardiovascular disease [24], high blood pressure (hypertension) [25], vision loss (diabetic retinopathy) [26], and kidney disease [27]. T2D results in chronically high blood glucose levels due to the inability of the pancreas to produce enough insulin to adequately regulate glucose levels [28,29]. Insulin is a peptide hormone produced by beta cells located in the pancreatic islets, where its primary function is to reduce the concentration of glucose in the bloodstream in order to maintain normal metabolic processes [30]. Since T2D development is associated with dysfunction in beta cells and insulin resistance [31], many studies have focused on the insulin signal transduction pathway [32,33,34], some of which have identified dysregulated lncRNAs, indicating their potential regulatory role in insulin signal transduction [35,36,37,38]. Although lncRNAs have been implicated in the pathogenesis of T2D [35,39,40,41], less than two dozen lncRNAs have been, to our knowledge, functionally studied (Table 1).

One complication of T2D is gastroparesis, a chronic condition in which the movement of food from the stomach to the small intestine is slowed or disrupted, leading to symptoms such as nausea, vomiting, and abdominal pain [62]. Diabetic gastroparesis is a type of gastroparesis that can occur in both type I and II diabetes, and accounts for approximately one third of all gastroparesis cases [63]. Grover et al. investigated cellular changes occurring in both diabetic and idiopathic gastroparesis by comparing full-thickness gastric body biopsies from patients with one of these two conditions to control subjects. They identified dysregulation in the innate immune system as a key feature of both diabetic and idiopathic gastroparesis [64]. However, little is known about the dysregulation of lncRNAs in gastroparesis, and thus further research is needed to understand their potential roles in the pathology of this condition.

Here, we systematically re-examined RNA-seq data from three published studies relating to diabetes (GSE164416, GSE115601, and GSE175988) with the aim to identify and annotate T2D-related lncRNA genes. Since very few lncRNAs have been functionally studied, we performed loss-of-function experiments of the lncRNA *USP30-AS1*, which we identified as dysregulated in T2D, using an in vitro model of pro-inflammatory macrophage activation. Finally, to facilitate further research of T2D-related lncRNAs, we built a web application, T2DB (https://rebeccadistefano.shinyapps.io/T2DB/; accessed on 26 April 2023), to access expression profiles of protein-coding and lncRNA genes identified in the three studies above, as well as to provide annotations of the identified lncRNA genes, which we expect to be a valuable resource in future studies elucidating the roles of non-coding RNAs in diabetes.

## 2. Results

### 2.1. Many lncRNAs Are Differentially Expressed in T2D Patients

To better understand the expression changes of lncRNAs related to insulin regulation, we re-examined RNA-seq data obtained from surgically removed pancreatic tissue samples of 133 metabolically phenotyped pancreatectomized patients [65] [Gene Expression Omnibus (GEO) accession number, GSE164416]. These samples were collected from 41 patients with impaired glucose tolerance (IGT), 35 patients with type 3c diabetes (T3cD, also known as pancreatogenic diabetes [66]), 39 T2D patients, and 18 subjects without T2D (control).

The primary objective of the original study [65] was to perform a multi-omics analysis at both the RNA and protein levels to profile living donors undergoing pancreatectomy. This approach was contrasted with previous research on human pancreatic islets, which had primarily focused on samples from individuals with normal levels of blood glucose (normoglycemic) or diseased brain-dead donors. The main findings of this study included progressive but disharmonic remodeling of mature beta cells, as well as the identification of gene co-expression modules and lipids that are associated with hemoglobin A1c (HbA1c) levels. However, the study did not analyze lncRNAs.

We conducted a differential expression analysis between the control group and each of the T2D, T3cD, and IGT patient groups, correcting for patient age, sex and BMI in accordance with the original study. This analysis revealed hundreds of up-regulated genes in T2D and T3cD patients compared to the control subjects, and fewer down-regulated genes (Figure 1A, Appendix A). In contrast, IGT patients exhibited minimal changes in gene expression relative to the control subjects, with four up-regulated protein-coding genes (myosin heavy chain 11 (MYH11), marker of proliferation Ki-67 (MKI67), apelin receptor (APLNR), and peptidase inhibitor 16 (PI16)) and one down-regulated lncRNA (the novel transcript ENSG00000285205). This observation is consistent with the understanding that IGT patients have not yet developed T2D, as their blood glucose levels are not high enough to warrant a T2D diagnosis [67].

Focusing on the up-regulated gene set, we found that 90 protein-coding and 8 lncRNA genes were shared between the T2D and T3cD comparisons (Appendix A). In contrast, there are 41 lncRNA and 222 protein-coding genes exclusively up-regulated in T2D (Figure 1B). The up-regulated protein-coding genes in the T2D group showed enrichment in Gene Ontology (GO) terms related to extracellular matrix organization, collagen biosynthetic process, and immune response (Figure 1C). The Kyoto Encyclopedia of Genes and Genomes (KEGG) terms reflecting protein digestion and absorption were also enriched, as well as Staphylococcus aureus infection (Figure 1D). To explore lncRNA genes dysregulated in T2D, we focused on the genes most correlated in their expressions to the up-regulated set of lncRNA genes and conducted a KEGG enrichment analysis. This analysis also showed protein digest and absorption, suggesting that the top correlating genes may be functionally related to the lncRNA genes themselves (Figure 1E). Interestingly, the most significantly up-regulated lncRNA in T2D patients, ENSG00000284653, was also up-regulated in T3cD patients (Appendix A). According to information from the recently updated LncBook 2.0 [68] (https://ngdc.cncb.ac.cn/lncbook/gene?geneid=HSALNG0001300; accessed on 26 April 2023), this intergenic lncRNA gene is conserved with mouse (the 50th quantile) and contains binding sites to the miRNAs hsa-miR-6743-5p and hsa-miR-6813-5p (Figure 1F). The most correlating genes to this lncRNA gene include Fas apoptotic inhibitory molecule 2 (FAIM2) (Figure 1F), whose polymorphism (dbSNP ID, rs7138803; https://www.ncbi.nlm.nih.gov/snp/rs7138803, accessed on 26 April 2023) is associated with obesity and heart rate in T2D subjects [69]. Taken together, these results warrant further functional and mechanistic studies of lncRNA genes in pancreas.

### 2.2. Diabetic Gastroparesis Results in Significant Down-Regulation in Gene Expressions

Next, we performed a similar analysis on a related dataset from Glover et al. [64] based on RNA-seq data using full thickness gastric body biopsies from seven diabetic gastroparetic (DG), seven diabetic non-gastroparetic control (DC), five idiopathic gastroparetic and seven non-diabetic non-gastroparetic control subjects (control) (GEO accession number, GSE115601). In total, 97 up- and 347 down-regulated genes in DG compared to the control subjects were identified (Figure 2A, Appendix A). On the other hand, only one up- (myosin heavy chain 2 (MYH2)) and two down-regulated protein-coding genes (Cbp/p300 interacting transactivator with Glu/Asp rich carboxy-terminal domain 2 (CITED2) and periplakin (PPL)) were identified in DC compared to the control subjects (Appendix A). Interestingly, MYH2 [70] and CITED2 [71,72,73] are known to be dysregulated in T2D, indicating that, unless the patients suffer from gastroparesis, the underlying diabetic condition does not show differences in gene expression patterns in gastric body biopsies. Among differentially expressed genes in DG compared to the control subjects, 41 up- and 60 down-regulated lncRNA genes were included (Appendix A), suggesting that the expression of lncRNA genes is altered in diabetic gastroparetic patients.

We next compared RNA-seq data between diabetic patients with or without gastroparesis, detecting in total 82 up- and 169 down-regulated genes, including 14 up- and 57 down-regulated lncRNA genes (Figure 2A, Appendix A), suggesting that gastroparesis results in more down-regulated genes in diabetic patients. Focusing on the down-regulated gene set only, we conducted a GO analysis, which revealed enriched terms related to immune and inflammatory responses as well as gene regulation (Figure 2B). Moreover, a KEGG pathway analysis indicated the enrichment of TNF, IL-17, JAK-STAT, and FoxO signaling pathways (Figure 2C). We repeated a similar KEGG pathway analysis, but involving genes most significantly correlated in their expression to the down-regulated lncRNA gene set (Figure 2D), revealing similar terms to when we directly analyzed the protein-coding genes. These results suggest that differentially expressed lncRNA genes might be involved in immune responses through the same signaling pathways as the genes they regulate. However, further functional studies are necessary to understand the biological roles of these differentially expressed lncRNA genes.

### 2.3. Many lncRNAs Are Differentially Expressed in Cultured Macrophages from T2D Patients

As demonstrated in the above studies [64,65] and in our re-examination of these datasets, it is clear that inflammatory cells significantly contribute to the pathogenesis and progression of T2D and its associated complications. To investigate this point further, Edgar et al. isolated human peripheral blood mononuclear cells (PBMC) from both T2D patients and control subjects without diabetes, who were matched for age and body mass index [74]. These cells were treated with the Toll-like receptor ligand lipopolysaccharide (LPS) and interferon-γ (IFN-γ) to induce inflammatory macrophages. Then, RNA-seq was performed to compare their gene expression profiles to the cells under the basal condition (10 unstimulated and 10 stimulated cells). The authors identified an enrichment in inflammatory response, IL-6 JAK STAT3 signaling, IL-α/IL-β signaling, Toll-like receptor 1/2 cascade, reactive oxygen species, and reactive nitrogen species production in stimulated diabetic cells. Here, we re-examined this dataset (GEO accession number, GSE175988) specifically focusing on lncRNA genes.

When comparing stimulated and unstimulated cells, a greater number of differentially expressed genes in PBMC isolated from T2D patients compared to control subjects were identified (Figure 3A), in accordance with the original publication [74]. Although some differentially expressed genes were shared between T2D and control samples (Figure 3B), 181 up- and 179 down-regulated genes were differentially regulated specifically in T2D compared to the control samples, including 35 up- and 22 down-regulated lncRNA genes (Appendix A). Among these up-regulated lncRNA genes, only the following nine have been previously studied: (1) ENSG00000232855 (novel transcript) [75]; (2) ENSG00000272512 (novel transcript) [76]; (3) KCNJ2 antisense RNA 1 (KCNJ2-AS1) [77,78,79]; (4) KLHDC7B divergent transcript (KLHDC7B-DT) [80,81,82]; (5) long intergenic non-protein-coding RNA 2015 (LINC02015) [83,84]; (6) long intergenic non-protein-coding RNA 2381 (LINC02381) [85,86,87,88,89,90,91,92,93]; (7) long intergenic non-protein-coding RNA 2541 (LINC02541) [94]; (8) serpin family B member 9 pseudogene 1 (SERPINB9P1) [95,96,97]; and (9) USP30 antisense RNA 1 (USP30-AS1) [98,99,100]. However, only the functions of the lncRNA genes LINC02381 and USP30_AS1 are known, with the former known to bind transcription factors [85], RNA-binding proteins [86], and miRNAs [87,88,89,90,91,92,93]; while USP30-AS1 acts as a miRNA sponge [98,99] and is involved in the cis-regulation of the nearby protein-coding genes USP30 and ANKRD13A [100]. Neither LINC02381 nor USP30-AS1 are known to be involved in T2D, which calls for further investigation.

USP30-AS1 is a lncRNA gene with one transcript, ENST00000478808.4, with two exons, located on Chromosome 12: 109,052,344-109,053,986 (reverse strand). To understand more about this lncRNA gene, we performed additional in-silico analysis, finding it to be most expressed in Epstein–Barr virus (EBV)-transformed lymphocytes (Appendix A), in support of its presence in PBMC. Upon correlating the CPM expression of USP30-AS1 with other genes, phospholipase A and acyltransferase 4 (PLAAT4) and serine/threonine kinase 26 (STK26) emerged as the most significant (Figure 3C). Moreover, the protein-coding genes JAZF zinc finger 1 (JAZF1) and X-box binding protein 1 (XBP1) were identified among the top correlated (FDR < 0.00001) and have been associated with the development and progression of T2D [101,102]. Lastly, the set of protein-coding genes demonstrating significant correlation with this specific lncRNA gene were enriched in the tumor necrosis factor (TNF) signaling pathway and necroptosis (Figure 3D). 

### 2.4. Loss-of-Function Experiments in Polarized Macrophages

As discussed in the previous section, none of the identified lncRNAs in PBMC from T2D patients have been functionally investigated in the context of T2D. To bridge this knowledge gap, we first evaluated the expression changes of selected T2D-related lncRNAs in an in vitro model of pro-inflammatory macrophage activation, by stimulating the human leukemia monocytic cell line, THP-1, to macrophage-like cells (M (-)) and then to pro-inflammatory macrophage-like cells (M (IFN-γ/LPS)), as described in the Materials and Methods Section (Figure 4A). Upon assessing the expression levels of the selected five T2D-related lncRNAs (ENSG00000272512, ENSG00000285040, ENSG00000289327, USP30-AS1, and KCNJ2-AS1) (Figure 4B), all of them were found to be highly up-regulated in pro-inflammatory cells compared to M (-) cells, although only USP30-AS1 exhibited statistically significant differences.

To understand its functional importance in macrophages, USP30-AS1 was silenced using siRNA (Figure 4C). Compared to the siRNA against scrambled sequence (siScr; control siRNA), silencing of USP30-AS1 resulted in down-regulation of pro-inflammatory marker genes, suggesting that USP30-AS1 is functionally important for the activation of pro-inflammatory macrophage-like cells.

As the name of the lncRNA gene suggests, USP30-AS1 overlaps with the nearby protein-coding gene, ubiquitin specific peptidase 30 (USP30), in an antisense direction. USP30 encodes for a deubiquitinating enzyme that is localized in the mitochondrial outer membrane and peroxisomes [103]. The functions of USP30 include PINK1/Parkin-mediated mitophagy, pexophagy, and BAX/BAK-dependent apoptosis, although nothing is known about its role in macrophages. Given that cis-regulation of the expression of a nearby protein-coding gene by a lncRNA is a common mechanism [104] and a previous study [100] reports that USP30-AS1 binds ASH2-like, histone lysine methyltransferase complex subunit (ASH2L) to regulate the promoter activity of USP30, we examined the expression change of USP30 upon silencing of USP30-AS1. Unlike the previous study [100], no significant change in the expression of USP30 was observed upon silencing of USP30-AS1 (Figure 4D), although its expression was up-regulated 796-fold upon activation of pro-inflammatory macrophage-like cells (Figure 4E). Taken together, the phenotypes observed here upon silencing of USP30-AS1 are independent of its cis-regulation of USP30. Thus, further mechanistic studies are required to elucidate the mechanism of action of UPS30-AS1 to the activation of macrophages.

### 2.5. The Web Application, T2DB, for Expression Analysis of Protein-Coding and lncRNA Genes

To facilitate further research on lncRNAs dysregulated in T2D, we built a web application called T2DB (https://rebeccadistefano.shinyapps.io/T2DB/; accessed on 26 April 2023) (Figure 5A). T2DB has four primary pages: (1) Explore; (2) LncRNA; (3) Download; and (4) Documentation. The Explore page shows an interactive Result Table (Figure 5B) displaying the results of the differential expression analysis for the study and comparison selected by the user. The right-hand side of the Explore page is divided into the following five tabs: (1) Volcano Plot; (2) Heatmap; (3) GO Analysis; (4) Pathway Analysis; and (5) Comparisons Intersection, which are related to the results shown in the Results Table. The table and the plot are linked so that selecting a row in the table will be reflected on the plot, in which the selected gene will be highlighted. A summary table displaying the number of up- and down-regulated genes (both protein-coding and lncRNA genes) is also displayed on this tab, which reactively changes as the user alters the threshold values of fold change in logarithm of base two scale (logFC) and FDR. These thresholds also affect the results displayed on the Volcano plot, Heatmap, GO Analysis, Pathway Analysis, and Comparisons Intersection tabs. The heatmap tab displays a heatmap, allowing the user to visualize the expression of the selected genes (Figure 5C). Gene Ontology enrichment analysis can be performed with FDR threshold values selected for multiple testing correction method and using the selected list of up- or down-regulated genes. The results are then visualized as a Manhattan plot and as a table. Only the results related to the Gene Ontology categories (Molecular Functions, Biological Process, and Cellular Components) are included. The Pathway Analysis tab displays the results of the KEGG pathway over-representation analysis performed using the selected list of up- or down-regulated genes. The results are visualized as a dot plot. Finally, the up- and down-regulated genes are compared across the different comparisons of one study and visualized in the Comparisons Intersection tab (Figure 5D).

The “lncRNAs” page allows the users to further examine the identified differentially expressed lncRNA genes (|log_2_FC| > 1 and FDR < 0.05) in each study and comparison through the lncRNA Table. This includes information about the conservation state of the lncRNA, along with its nearest protein-coding gene, and its most correlated gene in its expression using Pearson correlation. Finally, by selecting a row, the users can obtain further information about the miRNA associated with the selected lncRNA gene, along with the related Genome-wide association studies (GWAS) terms, based on the database LncBook 2.0 [68]. These are displayed on the right-hand side of the page. All the available data can be downloaded from the Download page so that further processing of the analyzed data is possible. Finally, the Documentation page provides information on the datasets used in the study and instructions on how to use the web application.

## 3. Discussion

In this study, we have re-examined RNA-seq data from three studies involving diabetes patients [64,65,74]. Our primary aim was to identify and analyze lncRNA genes dysregulated in T2D compared to control patients, a task not carried out in the original studies. We also built a highly informative wed database, T2DB, which provides a one-stop-shop for expression profiling of both protein-coding and lncRNA genes in T2D patients compared to control patients, as well as annotations for the identified dysregulated lncRNA genes. With respect to our re-analysis of protein-coding genes, we note that, while the original studies conducted differential analysis on these datasets, there are three good reasons for also including them in our study. First, lncRNA genes should not be analyzed in isolation to protein-coding genes, given their highly intertwined regulatory roles. Second, the choice of bioinformatic tools, such as mapping and differential expression programs, can make a large difference to results. Thus, for consistency purposes, it is important to apply the same tools for both lncRNA and protein-coding genes simultaneously in analyses. Third, the original authors did not gather their data into an easily accessible and interactive database, as we have done. As an extra comment, it is important to re-examine these kinds of data to provide confidence in the results from the original studies.

In our study, we have found that many genes, including lncRNA genes, are dysregulated in T2D patients compared to the healthy donors or subjects without T2D, and that the gene expression changes in T2D patients are augmented when the T2D progresses to other health complications (e.g., gastroparesis). Importantly, our results based on protein-coding gene analysis generally concur with those obtained in the original studies, although we have noted that many of our identified differentially expressed lncRNA genes have to date not been studied functionally or mechanistically. Interestingly, however, two of the lncRNA genes identified as dysregulated in diabetic gastroparetic were validated by the studies listed in Table 1; specifically, *NEAT1* was found to be down-regulated in DG patients compared to the control subjects, and *DNM3OS* was found to be up-regulated in DG patients compared to DC patients. Thus, our study provides a currently unappreciated connection between lncRNA dysregulation and the T2D complication known as diabetic gastroparesis.

Furthermore, our analyses based on correlating the expression of lncRNA genes to protein-coding genes resulted in the enrichment of pathways that are highly similar to those pathways identified based on the dysregulated protein-coding gene. Thus, it is likely that there is a close network-like dependency between lncRNA and protein-coding genes, which has become dysregulated in T2D, and these connections can be observed in the subsequent expression patterns. This is further supported by the fact that many of the closely correlating protein-coding genes have themselves been previously implicated in diabetes, such as *FAIM2*, *JAZF1*, and *XBP1*, as well as others whose connections likely remain uncharacterized.

One lncRNA gene, *USP30-AS1*, which we have found to be up-regulated in the stimulated cells of PBMC of T2D patients, has been analyzed previously, although not in the context of T2D. In this study, we have found that *USP30-AS1* influences the activation of macrophage-like cells. Interestingly, *USP30-AS1* has been implicated to be a potential prognostic risk factor for cancer and to play a role in cell death (e.g., apoptosis, autophagy, ferroptosis, and necroptosis) [105,106,107,108,109,110,111,112,113,114,115,116,117,118,119,120,121,122,123,124]. Furthermore, *USP30-AS1* promotes cell viability and inhibits apoptosis in acute myeloid leukemia cell lines, including THP-1 [100]. Yet, its functional role in macrophages is unknown. Here, we provide, for the first time, the potential function of *USP30-AS1* influencing the activation of pro-inflammatory macrophages. Finally, to disseminate the findings of this study to the broader audience, we built a web database, T2DB. We expect this database to be of broad interest to researchers looking into the molecular basis of T2D, and particularly in the context of non-coding RNAs. To facilitate this, we have offered flexibility to the user in terms of the parameters for selecting and analyzing differentially expressed genes. In addition, the annotations we have provided on the lncRNA genes identified in this study will be useful in helping researchers to prioritize candidates for potential functional studies.

With respect to limitations, we note that the RNA-seq data analyzed in this study come from different laboratories and are based on different protocols. In addition, all RNA-seq data are based on poly-A sequencing, so RNAs without poly A tails are not included in this analysis. Although we mitigated potential issues related to variations in RNA-seq data generation by analyzing each study separately and by comparing only the differentially expressed genes using a similar analysis pipeline, it is evident that we underestimated the number of T2D-related lncRNA genes, since more than half of lncRNA genes do not have poly A tails [125]. Furthermore, the functional data provided are of in vitro assay, which do not reflect the in vivo situation. The difficulty in performing experiments in immune cells (e.g., macrophages) is that even freshly isolated immune cells from an individual must be cultured to perform experiments. One way to perform in vivo experiments is to use model organisms (e.g., mice, zebrafish). However, most lncRNA genes are species-specific [126] and therefore it is not feasible to perform in vivo experiments for most human lncRNA genes as in our case here.

## 4. Materials and Methods

### 4.1. RNA-Seq Data Analysis and Visualization

As previously carried out [127,128], the raw RNA-seq data were downloaded from the Sequence Read Archive (SRA) database using SRA Toolkit [69]. After conversion of the .sra files to FASTQ files, the data were preprocessed with fastp [129] (version 0.21.0) using default settings to perform quality control, trimming of adapters, filtering by quality, and read pruning. To map the trimmed sequencing reads to the reference genome (GRCh38.107), STAR [130] (version 2.5.0a) was used. Using the R package, edgeR [131] (version 3.30.3), CPM values and differentially expressed genes were calculated and derived, respectively. With respect to GSE164416, we included co-variates for BMI, sex and age in accordence with the original study. FDR-adjusted *p*-values calculated by the Benjamini–Hochberg method were used for further analysis. For all comparisons, the threshold values of |log_2_FC| > 1 and FDR < 0.05 were used as criteria for calling differentially expressed genes.

To draw volcano plots, the R-package ggplot2 [132] was used. To derive overlapping genes, http://bioinformatics.psb.ugent.be/webtools/Venn/ (accessed on 7 October 2022) was used. To identify enriched GO terms and KEGG pathways, Database for Annotation, Visualization, and Integrated Discovery (DAVID) (version v2022q3) [133,134] was used for GOTERM_BP_DIRECT and KEGG_PATHWAY categories, respectively, based on *p*-values computed by DAVID by applying Fisher’s Exact test. The top enriched GO terms and KEGG pathways were selected based on -logarithm of base 2 of *p*-values. Correlation analysis between lncRNA and other genes was based on Pearson’s correlation test and CPM values. Top correlating protein-coding genes to either a single lncRNA gene or a set of genes were selected based on FDR < 0.00001. LncRNA genes identified as dysregulated were annotated using LncBook 2.0 as follows. First, lncRNA coordinates from the gene transfer format (GTF) file based on GRCh38.107 were overlapped with coordinates from LncBook 2.0 (LncBookv2.0_GENCODEv34_GRCh38.gtf) to obtain the relevant LncBook ID. These ids were compared with LncBook 2.0 annotations on conservation (conservation_LncBook2.0.csv), miRNA-binding sites (lncrna_mirna_miRandaAndTargetScanAndRNAhybrid_LncBook2.0.csv), and GWAS traits (variation_LncBook2.0.csv). We also included annotations on the nearest gene using the R package GenomicRanges [135] and the top correlation gene according to Pearson’s correlation test.

### 4.2. Cell Culture and Treatments

The human leukemia monocytic cell line, THP-1 (LGC Standards GmbH (Wesel, Germany), #ATCC-TIB-202; Lot #70043382), was cultured in the growth medium containing RPMI 1640 Medium (Thermo Fisher Scientific (Roskilde, Denmark), #21875091) supplemented with 10% fetal bovine serum (Sigma-Aldrich (Søborg, Denmark), #F4135), 1% L-Glutamine solution (Sigma-Aldrich, #G7513), and 1% Penicillin-Streptomycin (Sigma-Aldrich, #P4333) at 37 °C with 5% CO_2_. To differentiate THP-1 cells into macrophage-like cells (M (-)), the cells were incubated in the growth medium containing 100 nM of phorbol 12-myristate 13-acetate (PMA; Sigma-Aldrich, #P8139-1MG) for 3 days. To induce pro-inflammatory macrophage activation, M (-) were incubated with the growth medium supplemented with 20 ng/mL of recombinant human interferon-γ (IFN-γ; Cell Signaling, #80385) and 250 ng/mL of eBioscience Lipopolysaccharide (LPS) Solution (500X; 2.5 mg/mL) (Thermo Fisher Scientific, #00-4976-93) for 48 h to derive activated macrophage-like cells (M (IFN-γ/LPS)).

To silence the USP30-AS1, the MISSION siRNAs (Sigma-Aldrich) were used: siUSP30-AS1.1 sense, CUUAGGCUCCAUUCAUAAAdTdT, antisense, UUUAUGAAUGGAGCCUAAGdTdT; and siUSP30-AS1.2 sense, GCUUAGGCUCCAUUCAUAAdTdT, antisense, UUAUGAAUGGAGCCUAAGCdTdT. Mission Negative control SIC-002, confidential sequence (Sigma-Aldrich), was used as control. Thirty minutes after inducing the activation of pro-inflammatory macrophage-like cells, transient siRNA transfection (10 nM final concentration) was carried out using RNAiMax (Thermo Fisher Scientific (Roskilde, Denmark) according to the manufacturer’s protocol. The samples were collected 48 h after the transfection of siRNAs for the isolation of total RNA.

### 4.3. Isolation of Total RNA and RT-PCR

To isolate and purify the total RNA, the TRIzol Reagent (Thermo Fisher Scientific, Roskilde, Denmark, #15596018) was used following the manufacturer’s protocol [128]. To synthesize the first-strand complementary DNA (cDNA) from 1 μg of total RNA for each sample, SuperScript IV VILO Master Mix with the ezDNase Enzyme (Thermo Fisher Scientific, #11766500) was used to digest the genomic DNA and reverse transcribe the total RNA. Then, the first-strand cDNA was diluted with DNase/RNase-free water to the concentration of 1 ng/μL. A quantitative reverse transcription polymerase chain reaction (qRT-PCR) reaction was performed using 1 ng of cDNA template per reaction with PowerUp SYBR Green Master Mix (Thermo Fisher Scientific, #A25777) via the QuantStudio 6 Flex Real-Time PCR System (Thermo Fisher Scientific). The annealing temperature was set to 60 °C. To normalize the data, relative fold expression was calculated by 2-DDCt using ribosomal protein lateral stalk subunit P0 (*PRLP0*) as an internal control. The primer pairs for markers of activated macrophages were previously published by others [136]. The primer pairs for lncRNAs were designed using Primer3 (http://bioinfo.ut.ee/primer3-0.4.0/; accessed on 6 September 2022) [78] and in silico validated with the UCSC In-Silico PCR tool (https://genome.ucsc.edu/cgi-bin/hgPcr; accessed on 6 September 2022) before extensive testing by the conventional RT-PCR reaction followed by running the PCR product on an agarose gel to examine for a single band of the expected size for each primer pair. The primer sequences are provided in Appendix A.

### 4.4. T2DB Web Application

The T2DB web application is based on the R package Shiny [137]. The app has four primary pages: (1) Explore; (2) *LncRNA*; (3) Download; and (4) Documentation. The Explore page displays an interactive Result Table, which is created with the R package DT (https://github.com/rstudio/DT; accessed on 21 March 2023). The right-hand side of the Explore page is divided into the following five tabs: (1) Volcano Plot created with the R package ggplot2 [132]; (2) Heatmap using the function pheatmap from the R package, ComplexHeatmaps [138]; (3) GO Analysis using the R package, gprofiler2 [139], and displayed as a Manhattan plot using the gostplot function from gprofiler2 and as a table rendered using the R package DT; (4) Pathway Analysis using the enrichKEGG function from the R package, clusterProfiler [140,141], and visualized as a dot plot created with the R package, enrichplot [142]; and (5) Comparisons Intersection using the upset function of the R package, UpSetR [143].

The lncRNAs page displays an interactive lncRNA Table rendered using the R package DT showing the differentially expressed lncRNA genes (|log_2_FC| > 1 and FDR < 0.05) for the selected study and comparison. This can be downloaded in the format of tab-separated values (.tsv), along with the information displayed on the right-hand side of the page.

The Download page displays an interactive table showing the data for the selected study and rendered using the R package DT. The selected study can then be downloaded in either the format of comma-separated values (.csv) or .tsv. The Documentation page provides information on the datasets used in the study and instructions on how to use the web application.

All code used to generate T2DB is available in the GitHub repository: https://github.com/Reb08/T2DB (accessed on 26 April 2023). T2DB is freely available without password from https://rebeccadistefano.shinyapps.io/T2DB/ (accessed on 26 April 2023).

### 4.5. Statistics

Data are presented as the mean ± S.E.M. Two-sample, two-tail, heteroscedastic Student’s *t*-test was performed to calculate a *p*-value via Microsoft Excel.

## Figures and Tables

**Figure 1 ncrna-09-00030-f001:**
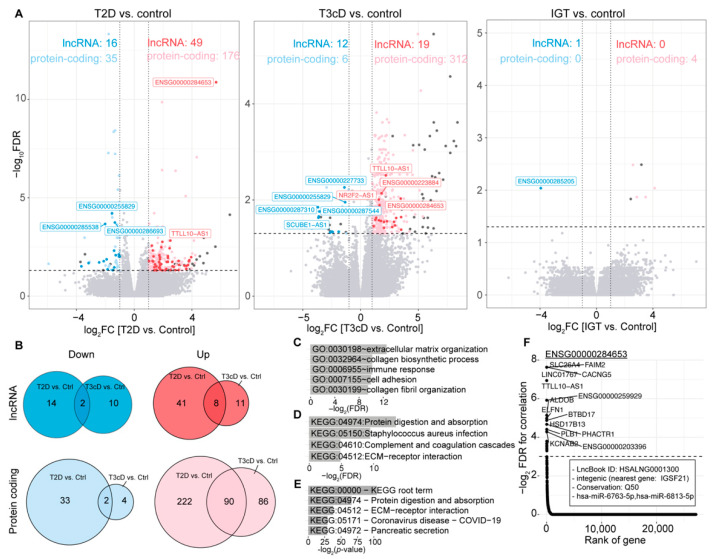
RNA-seq data analysis of surgical pancreatic tissue samples from 41 patients with impaired glucose tolerance (IGT), 35 patients with type 3c diabetes (T3cD), 39 T2D patients, and 18 subjects without T2D (control). (**A**) Volcano plots. The threshold values of absolute value of logarithm of base 2-in fold change (|log_2_FC|) of 1 and false discovery rate-adjusted *p*-value (FDR) < 0.05 were applied. Red and pink dots represent up-regulated lncRNA and protein-coding genes, respectively, and blue and light-blue dots represent down-regulated lncRNA and protein-coding genes, respectively, compared to the control samples in all cases. (**B**) Venn diagrams depicting up- and down-regulated lncRNA and protein-coding genes comparing T2D and T3cD patients against the control subjects. (**C**–**E**) Enriched (**C**) GO terms for up-regulated T2D-related genes (top 5 only); (**D**) KEGG pathways; and (**E**) KEGG pathways for the most significantly correlating protein-coding genes to the set of up-regulated lncRNA genes (top 5 only). (**F**) Correlation of the counts per million (CPM) values between the lncRNA *ENSG00000284653* and all expressed protein-coding genes, with the top correlating genes depicted. The box indicates LncBook 2.0-collected annotations about this lncRNA gene, including its type, conservation percentile, and predicted miRNA binding sites.

**Figure 2 ncrna-09-00030-f002:**
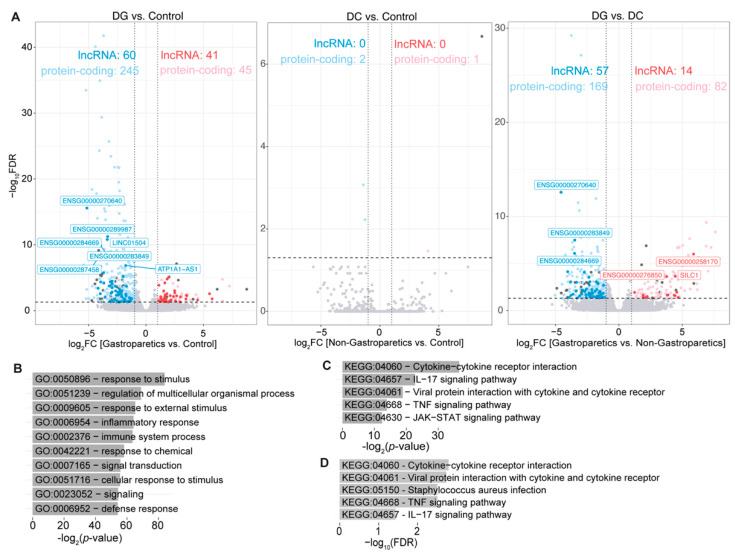
RNA-seq data analysis of full thickness gastric body biopsies from seven DG, seven DC, and seven control subjects. (**A**) Volcano plots displaying threshold values of |log_2_FC| > 1. Red and pink dots represent up-regulated lncRNA and protein-coding genes, respectively, while blue and light blue dots represent down-regulated lncRNA and protein-coding genes, respectively, compared to the control samples. (**B**,**C**) Top (**B**) 10 enriched GO terms; and (**C**) Top 5 KEGG pathways for down-regulated protein-coding genes. (**D**) Top five KEGG pathways of protein-coding genes correlating to the set of down-regulated lncRNA genes (FDR < 0.00001).

**Figure 3 ncrna-09-00030-f003:**
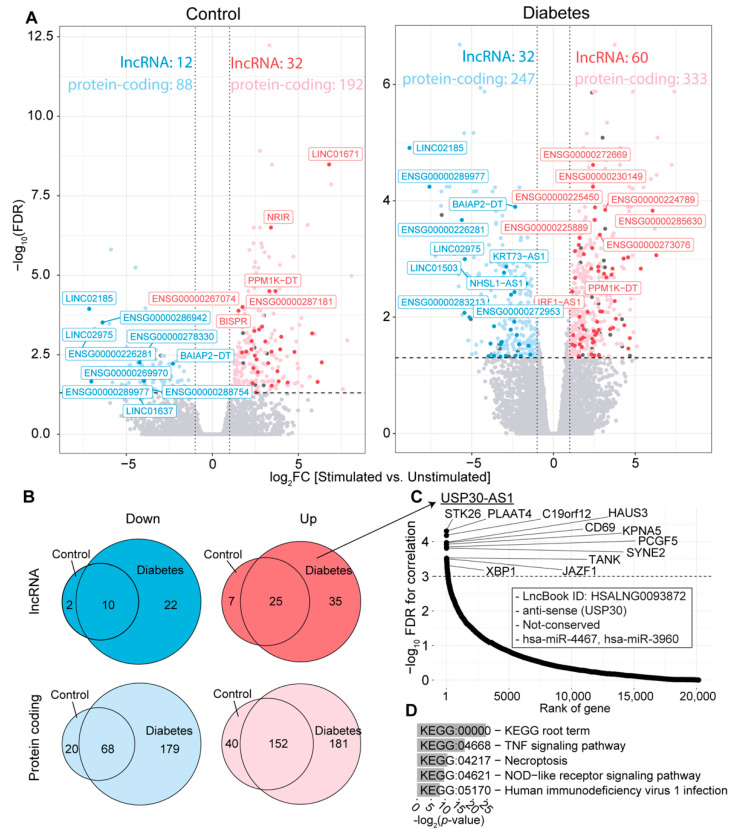
RNA-seq data analysis of PBMC isolated from four control subjects and six T2D patients. All cells were stimulated with LPS and IFN-γ, which are marked as “stimulated”, and compared to the unstimulated samples. (**A**) Volcano plots including thresholds of |log_2_FC| > 1 and FDR < 0.05. Red and pink dots represent up-regulated lncRNA and protein-coding genes, respectively, while blue and light blue dots represent down-regulated lncRNA and protein-coding genes, respectively, compared to the unstimulated samples in all cases. (**B**) Venn diagrams of up- and down-regulated protein-coding and lncRNA genes shared between the two comparisons. (**C**) In silico analysis of the lncRNA gene *USP30-AS1*. The ranked correlations between the expression of *USP30-AS1* and all other genes are shown along with the information from LncBook 2.0. (**D**) Top 5 KEGG pathway analysis of the top correlating genes to *USP30-AS1* (FDR < 0.00001).

**Figure 4 ncrna-09-00030-f004:**
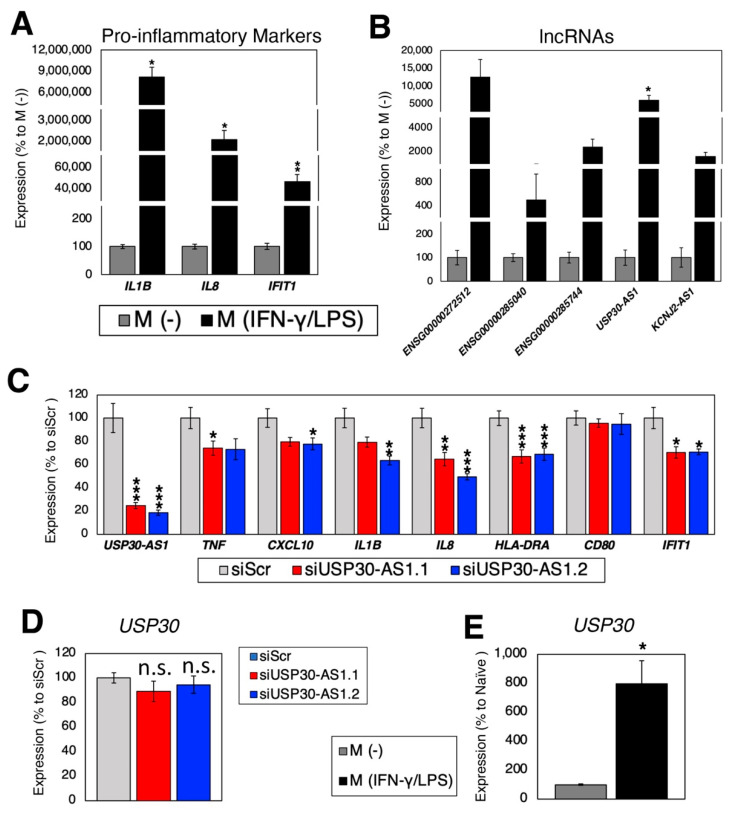
Expression and functional analysis in macrophages. (**A**,**B**) Expressions of (**A**) pro-inflammatory marker genes; and (**B**) the selected lncRNAs. n = 6 and 4 biological replicates per M (-) and M (IFN-γ/LPS) cells, respectively. * (*p* < 0.05), ** (*p* < 0.01) and *** (*p* < 0.005). (**C**) Silencing of *USP30-AS1.* Compared to the control (siRNA against scrambled sequence (siScr)), two siRNAs against *USP30-AS1* (siUSP30-AS1.1 and siUSP30-AS1.2) resulted in down-regulation of pro-inflammatory marker genes. n = 6 biological replicates. (**D**,**E**) Expressions of the overlapping protein-coding gene, *USP30*. (**D**) Upon silencing of *USP30-AS1*. n = 6 biological replicates. n.s. stands for not significant. (**E**) Comparing M (-) and M (IFN-γ/LPS) cells. n = 6 and 4 biological replicates per M (-) and M (IFN-γ/LPS) cells, respectively.

**Figure 5 ncrna-09-00030-f005:**
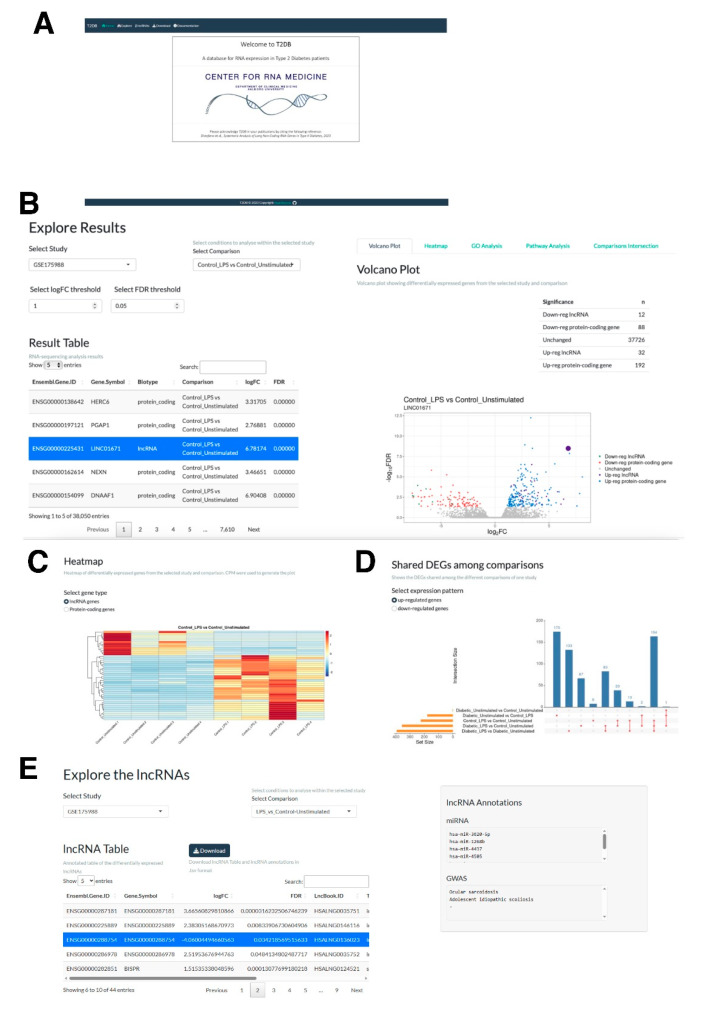
T2DB web database. (**A**) The top page of T2DB. (**B**) “Explore Results” tab and window. The users can dynamically explore each differentially expressed genes from the Result Table. (**C**) The differentially expressed genes can be visualized via a heat map. (**D**) As there are several conditions within each study, differentially expressed genes among different conditions to be compared can be visualized via the “Comparisons Intersection” Table. (**E**) “Explore the lncRNAs” tab and window. The users can dynamically explore each differentially expressed lncRNA gene in each study and comparison from the lncRNA Table.

**Table 1 ncrna-09-00030-t001:** List of functionally studied lncRNAs in relation to T2D.

Official Gene Name	Official Gene Symbol and Alias	Reference
ARAP1 antisense RNA 2	*ARAP1-AS2*	[42]
TRAF3IP2 antisense RNA 1	*TRAF3IP2-AS1*, also known as *βFaar*	[43]
cytochrome P450, family 4, subfamily b, polypeptide 1, pseudogene 1	*Cyp4b1-ps1*, also known as *CYP4B1-PS1-001*	[44]
DNM3 opposite strand/antisense RNA	*Dnm3os*	[45]
diabetes regulated anti-inflammatory lncRNA	*DRAIR*, also known as *CPEB2-AS*	[46]
growth arrest specific 5	*GAS5*	[47]
predicted gene 10768	*Gm10768*	[48]
H19 imprinted maternally expressed transcript	*H19*	[49]
KCNQ1 opposite strand/antisense transcript 1	*KCNQ1OT1*	[50,51]
tumor protein p53 pathway corepressor 1	*TP53COR1*, also known as *LINC-P21*	[52]
long intergenic non-protein-coding RNA 1619	*LINC01619*	[53]
predicted gene, 19689	*Gm19689*, also known as *lncSHGL*	[54]
LYPLAL1 divergent transcript	*LYPLAL1-DT*	[55]
myocardial infarction associated transcript	*MIAT*	[56]
nuclear paraspeckle assembly transcript 1	*NEAT1*	[57]
prostaglandin-endoperoxide synthase 2	*PTGS2*	[58]
small nucleolar RNA host gene 5	*SNHG5*	[59]
taurine up-regulated 1	*TUG1*	[60]
TCL1 upstream neural differentiation-associated RNA	*TUNAR*	[61]
ARAP1 antisense RNA 2	*ARAP1-AS2*	[42]
TRAF3IP2 antisense RNA 1	*TRAF3IP2-AS1*, also known as *βFaar*	[43]

## Data Availability

The Appendix A can be found on the GitHub repository: https://github.com/heartlncrna/Analysis_of_T2D_Studies (accessed on 1 May 2023). All codes used to generate T2DB are available on the GitHub repository: https://github.com/Reb08/T2DB (accessed on 26 April 2023).

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
