# Peer review of "T2DB: A Web Database for Long Non-Coding RNA Genes in Type II Diabetes"

_ncrna, 2023, doi:10.3390/ncrna9030030_

Round 1
Reviewer 1 Report
This manuscript by Distefano et al, aimed at investigating whether lncRNAs would be important as important elements for Type II diabetes (T2D). The authors used previous datasets to re-analyze taking into account the transcriptomic signal from the non-coding part of the genome, finding potentially useful elements strongly correlated with T2D for future studies. The article is well structured, the reading is clear and easy to follow and the results obtained are useful and suitable for further studies in in vitro assays to validate this data. After having carefully reviewed the manuscript as well as figures and supplemental data, I recommend the article for publication after solve several questions:
>Lines 21-23 This sentence feels redundant “increases… increasing”
>Lines 66-80 It would be recomendable to convert these lines into a table.
>Figure 1 I cannot follow the numbers: In fig. 1C there were included all genes (protein-coding, lncRNAs and others), which results in 184 (up) + 58 (down) = 242. However, in fig. 1D, it says that the GO analysis was made just with the protein coding genes (that makes sense), but the number of genes fits perfectly with the previous sum including altogether.
>Figure 1D Did the authors try to perform the GO ontology with the up/down lists separately?
>Lines 146-148 A better description and explanation of the relationship between Gastroparesis and Type II diabetes could be very helpful.
>Figure 2D and 2E Same question regarding separated list of up and down regulated genes.
Reviewer 2 Report
The authors performed secondary analyses of publicly available RNA-sequencing datasets from three different sources and representing two different conditions. The main stated objective of the investigation was to identify lncRNAs showing differential expression between individuals with T2D and healthy controls.
Overall, the manuscript is poorly organized, contains many grammatical errors, and presents only a superficial discussion of the findings. Furthermore, the rationale for the project is not strongly presented and the Results section contains information that would be more appropriate for the Introduction section. In addition to these impressions of the manuscript, the following concerns further weaken the impact of the work.
1. The statistical analyses did not adjust for any potential covariates, such as age, sex, BMI, ethnicity, duration of diabetes, etc, which could significantly affect the results. There was also no validation of findings using other publicly available datasets.
2. The authors indicate an intention to focus on lncRNA expression, yet, the Results section contains a significant amount of information related to differential expression of protein-coding genes. This information is redundant, as results of differences in protein-coding genes were published in the original works. What is the rationale for providing results of protein-coding gene expression? If there is a rationale for repeating these analyses, then results from the current work should be compared and contrasted with those from the published studies. For example, the differential expression of protein-coding genes in IGT reported here contrasts with the original work.
3. The analysis of T2D and gastroparesis and three different sources of RNA comes off as disjointed and unconnected. There is no cohesion among these different lines of thinking. Even with the analyses of RNA from pancreatic tissue and PBMCs from people with T2D, there wasn’t an attempt to look for differentially expressed lncRNAs that might be shared between the two datasets.
4. The rationale for functionally characterizing a lncRNA showing differential expression in PBMCs is not strong. It seems more fitting to characterize a lncRNA differentially expressed in pancreatic tissue as the focus is on T2D and lncRNAs in the pancreas are more likely to have a functional role than those circulating in PBMCs.
5. The Discussion section is superficial and does not address the implications of the results. For example, were any of the lncRNA previously associated with T2D replicated in this work?
6. The sample size from the PBMC analysis (Edgar et al) is not indicated. How many T2D and control samples were analyzed?
7. The sample sizes in the remaining datasets were modest, especially for the gastroparesis study. Results from such a small sample size are not convincing.
Round 2
Reviewer 2 Report
No additional comments.